# Capacitance–Voltage Fluctuation of Si_x_N_y_-Based Metal–Insulator–Metal Capacitor Due to Silane Surface Treatment

**DOI:** 10.3390/mi15101204

**Published:** 2024-09-28

**Authors:** Tae-Min Choi, Eun-Su Jung, Jin-Uk Yoo, Hwa-Rim Lee, Sung-Gyu Pyo

**Affiliations:** School of Integrative Engineering, Chung-Ang University, 84, Heukseok-ro, Dongjak-gu, Seoul 06974, Republic of Korea; c79411@gmail.com (T.-M.C.); eunsuj@cau.ac.kr (E.-S.J.); wlsdnr5771@naver.com (J.-U.Y.); ghkfla0725@naver.com (H.-R.L.)

**Keywords:** MIM, capacitors, metal–insulator–metal, electrical performance, Si_x_N_y_, cap density, VCC, TCC

## Abstract

In this study, we analyze metal–insulator–metal (MIM) capacitors with different thicknesses of SixNy film (650 Å, 500 Å, and 400 Å) and varying levels of film quality to improve their capacitance density. SixNy thicknesses of 650 Å, 500 Å, and 400 Å are used with four different conditions, designated as MIM (N content 1.49), NEWMIM (N content 28.1), DAMANIT (N content 1.43), and NIT (N content 0.30). We divide the C–V characteristics into two categories: voltage coefficient of capacitance (VCC) and temperature coefficient of capacitance (TCC). There was an overall increase in the VCC as the thickness of the SixNy film decreased, with some variation depending on the condition. However, the TCC did not vary significantly with thickness, only with condition. At the same thickness, the NIT condition yielded the highest capacitance density, while the MIM condition showed the lowest capacitance density. This difference was due to the actual thickness of the film and the variation in its k-value depending on the condition. The most influential factor for capacitance uniformity was the thickness uniformity of the SixNy film.

## 1. Introduction

A metal–insulator–metal (MIM) capacitor is an analog integrated circuit (IC) configuration device with the advantages of low electrode resistance and parasitic capacitance [1,2,3,4]. MIM capacitors have high charge mobility and burst power characteristics that make them excellent energy-storage devices and potential auxiliary power sources.

IM capacitors have been applied to ICs such as high-power microprocessor units and dynamic random-access memory. However, with the development of wireless communication, their application to radio frequency (RF) devices has been actively studied [5,6,7]. As current RF devices require high operating frequencies, MIM devices also require high capacitance per unit area [8,9,10,11].

According to this demand, the design of the structure, thin-film deposition method, selection of the bottom and top electrode materials, insulator material, thickness of the electrode and insulator, dielectric constant of the insulator, and crystal structure of the insulator must be considered in depth to produce a high-performance MIM capacitor with a high capacitance [12,13,14,15]. In the evaluation of MIM capacitors, it is important to conduct a comprehensive analysis considering factors such as capacitance density (CD), leakage current density, charge storage density, and dielectric breakdown strength [16,17,18,19].

The choice of insulator material is a crucial factor in capacitors, and the CD relies heavily on the dielectric constant and thickness of the insulator. According to Equation (1), the CD increases when the dielectric constant is higher, which is a natural property of the insulator, and when the thickness is lower [1,20]:(1)$$C=\frac{k\varepsilon_{0}A}{d}\to \frac{C}{A}=\frac{k\varepsilon_{0}}{d}$$
where *C* denotes the capacitance (F), *k* is the dielectric constant, *ε*_0_ is the permittivity of the vacuum (8.854 × 10^−12^/m), and d is the thickness of the insulator (m).

Because of the abovementioned reasons, it is generally necessary to introduce a high-k material to increase the dielectric constant [21,22], which entails considerable investment and time because it requires equipment, facilities, and source replacement. In addition, according to Natori et al., the relative permittivity of the material (k) decreases as the insulator thickness of the capacitor decreases, and k tends to decrease significantly for high-k materials [13,23]. Therefore, the introduction of high-k materials is subject to many limitations. However, if the thickness of the currently used medium-k dielectric material, Si_x_N_y_ (k = 7), can be reduced by considering the leakage aspect, the capacitance value can be increased without using a high-k material [24,25,26]. Yu et al. explained the performance of HfO_2_-based MIM capacitors deposited by the atomic layer deposition (ALD) method with respect to the thickness of the dielectric [27]. As the thickness of the HfO_2_ insulator layer decreased, the CD and voltage coefficient of capacitance (VCC) increased [28,29,30,31]. In practice, GaAs-based MIM capacitors have been used in the past. In fact, for GaAs-based MIM capacitors, Si_x_N_y_ is the most commonly applied material owing to its excellent electrical properties, compliant dielectric constant, high dielectric breakdown voltage, and low leakage current [28,32,33]. Moreover, the electrical properties can be improved by optimizing the deposition condition of Si_x_N_y_, which has the greatest effect on the electrical properties of MIM capacitors. Yota et al. confirmed that the stress, CD, breakdown voltage, and performance of MIM capacitors exhibited significant differences in each insulator layer, with a single layer or multiple layers of silicon nitride formed depending on the deposition conditions [34]. Therefore, the electrical properties of MIM capacitors can be improved by optimizing the deposition condition of Si_x_N_y_.

In this study, to develop an optimal condition for the deposition conditions of Si_x_N_y_ that improves the insulator properties of MIM capacitors and secures feasibility, we fabricated MIM capacitors with different thicknesses of Si_x_N_y_ and deposition conditions on M4 wiring and then evaluated the capacitance–voltage (C–V) characteristics, focusing on the evaluation of cap density uniformity, the dielectric temperature coefficient of capacitance (TCC), and the VCC.

## 2. Materials and Methods

Patterned 200 mm Si (100) wafers were used to measure the integration process steps. Several different cap dielectrics were investigated and deposited by PE-ALD. An Applied Materials MIRRA tool(Applied Materials, Gloucester, MA, USA) was used for the blanket. In this study, the density of MIM capacitors was considered to be 8 fF/m^2^, and the capacitors were fully integrated using the 0.15 m Al interconnect processes. The first single MIM capacitor was formed using metal 3 and metal 4 to minimize the effect of the parasitic coupling of the silicon substrates. In addition, the second single MIM capacitor was formed using metal 5 and metal 6. It is crucial for MIM capacitors to have a symmetric structure by having identical boundary conditions on both sides of the dielectric [8,35,36].

The bottom electrode of the MIM capacitor was prepared using Ti (100 Å)/Al–Cu (4500 Å)/Ti (50 Å)/TiN (600 Å) wiring, and the top electrode was prepared using TiN (1500 Å). The insulator was SixNy. The capacitor fabrication process was as follows: bottom electrode deposition → bottom electrode scrub → insulator deposition → top metal deposition → MIM PH → MIM → TOP METAL etching → ((CH_3_)4NOH:H_2_O) cleaning 1 → MIM asher → ((CH_3_)4NOH:H_2_O) cleaning 2 → insulator etching → ACT 935 (wet PR strip solution including amine) →UVAS. MIM ET was performed using the endpoint detection method.

The top electrode layer was connected to the upper metal layer through a dense matrix of vias. All wafers mentioned in this paper were passivated using PE-ALD nitride and were annealed below 450 °C. To improve the voltage linearity, Interface plasma treatment was administered before and after the PE-ALD dielectrics. Further details of the interface plasma treatment and thickness ratio in each stack layer are listed in Table 1. The blanket film characteristics of PE-ALD dielectrics were evaluated by using an ellipsometer at 673 nm to measure the thickness, refractive index, and uniformity of the PE-ALD dielectrics. A Hg probe was used to measure the dielectric constant, and the deposition rate was calculated according to the thickness slope as a function of the cycle times [4].

The C–V characteristics were measured manually using an LCR meter (HP4284A, Agilent, Santa Clara, CA, USA) under the conditions given in Table 2. The VRDB was performed based on the JESD35-A standard. The thickness analysis of Si_x_N_y_ per condition was performed using transmission electron microscopy (CM200FEGTEM)/scanning transmission electron microscopy (STEMoperated at 300 keV with an energy-dispersive X-ray spectroscopy (EDS) SUTW-SiLi X-ray detector and a Gatan 666 parallel electron energy loss spectroscopy (PEELS) spectrometer (Philips, Eindhoven, Netherlands), and a focused ion beam (FIB). The via resistance and via chain yields were measured in dual-damascene structures. A wafer-level bias thermal stress (BTS) test was performed under different conditions to verify the effectiveness of the barrier layers. Failures were analyzed by scanning electron microscopy (SEM) and X-ray spectroscopy (EDX).(SIGMA, Carl Zeiss, Jena, Germany) 

## 3. Results and Discussion

To understand the wafer-wide trend in CD, PCM measurements were performed with the split conditions given in Table 1; the corresponding results for a 25 × 25 cap size are presented in Figure 1.

The CD varied depending on the split condition but was uniform within the wafer. For all conditions, the CD increased with decreasing thickness, and at the same thickness, it varied slightly between conditions. By examining the range of CD in relation to thickness, the following values were observed for thicknesses of 650, 500, and 400 Å, respectively: 0.983–1.1, 1.24–1.4, and 1.57–1.79 fF/μm^2^. The difference in capacitance densities between conditions at the same thickness can be considered to be the difference between the actual thickness of the Si_x_N_y_ film and the target thickness and the difference in the k-value of the deposited film by condition. To confirm this, 500 Å thick Si_x_N_y_ films deposited under the different conditions were analyzed by TEM; the corresponding results are shown in Figure 2.

Figure 2a–d depict the cross-sectional TEM images of Si_x_N_y_ films deposited under MIM, NEWMIM, DAMANIT, and NIT conditions, respectively, and Figure 2e shows a barplot of the thickness of Si_x_N_y_ films obtained from the TEM images. The NIT condition showed the lowest thickness, while the NEWMIM condition showed the highest thickness. For accurate analysis, the k-value was calculated after matching the TEM analysis die and the PCM measurement die; the corresponding results are presented in Table 3.

In this experiment, we compared the capacitance densities of four materials with the same thickness (500 Å) and found that NIT afforded the highest CD, followed by NEWMIM, DAMANIT, and MIM. However, we noticed that the deposited thickness did not follow this trend. This indicates that thickness alone is not the only factor that affects the CD. The difference in the k-value according to the condition also appears to play a significant role in determining the CD. In general, the k-value is influenced by two primary factors, namely, the macroscopic electric field and the dipole moment per unit volume, as given by Equation (2) [2]:(2)$$K=1+\frac{4\pi P}{E}$$
where *P* is the dipole moment per unit volume and *E* is the macroscopic electric field.

Because the *p*-value is dependent on the electronic polarizability, it is affected by the bond conformation and bond strength [21]. The Si_x_N_y_ films had different dipole moments due to the different values of Si-H/N-H (Table 4) depending on the deposition condition; therefore, the k-value was different for each condition.

To check the variation in CD according to capacitor size, the capacitor density by thickness and condition was measured for 10 × 10, 15 × 15, 20 × 20, 25 × 25, 30 × 30, and 50 × 50 μm^2^ samples; it is plotted in Figure 3. In this case, the CD according to size was taken as the average value within the wafer.

It can be observed that the CD decreases as the size of the capacitor increases, regardless of the thickness and condition of Si_x_N_y_. After a certain point, the CD remains constant. This phenomenon can be attributed to the effect of fringe capacitance due to the perimeter/area ratio and the variation in fringe impedance CD (FICD) with respect to size. The difference between capacitor sizes of 10 × 10 and 20 × 20 μm^2^ is more pronounced in the case of FICD variation, as it has a greater impact on smaller sizes [37].

In terms of device fabrication, the uniformity of the CD is closely related to the process capability index (Cp, Cpk), with Si_x_N_y_ thickness uniformity being the most important factor.

As depicted in Figure 4, although there are a few points that deviate from the linear trend, a proportional relationship exists between Si_x_N_y_ thickness nonuniformity and CD nonuniformity, with a slope of 1.04.

Improving the uniformity of Si_x_N_y_ thickness can result in an improvement in CD uniformity, which, in turn, can increase the values of Cp and Cpk on the device manufacturing side to 1.33 or above.

To measure the VCC, an index indicating the degree of change in capacitance with respect to voltage variations, measurements were taken at the top, center, and bottom of the wafers according to the thickness of Si_x_N_y_ and the conditions. The VCC is denoted by *V_cc_*_1_ (ppm/dV) and *V_cc_*_2_ (ppm/dV^2^), as expressed by Equation (3) [38]:(3)$$\frac{C\left(V\right)-C\left(0\right)}{C\left(0\right)}=V_{cc2}V^{2}+V_{cc1}V+C$$
where *C*(*V*) is the capacitance under variable voltage, *C*(0) is the capacitance at 0 V, *V_cc_*_1_ and *V_cc_*_2_ are the VCCs, and *C* is a constant value.

The VCC graph was plotted by performing a polynomial fit with the voltage on the *X*-axis and the normalized Δ*C* on the *y*-axis, as described in Equation (3). As an example, the VCC graph for the Si_x_N_y_ film processed with the NEWMIM condition at a thickness of 500 Å is depicted in Figure 5.

The VCC graph results for the Si_x_N_y_ films, which vary in thickness (650 Å, 500 Å, and 400 Å) and condition (MIM, NEWMIM, DAMANIT, and NIT), are summarized in Figure 6. Figure 6a displays the *V_cc_*_1_ values according to thickness, while Figure 6b illustrates the *V_cc_*_2_ values as a function of thickness. Both the *V_cc_*_1_ and *V_cc_*_2_ values showed an increasing trend as the thickness decreased, with the initial level and degree of increase varying according to the condition.

In the case of V_cc1_, all conditions showed values below 60 ppm/dV at 650 Ǻ, but they values increased as the thickness decreased, and only the MIM and NEWMIM conditions showed values over 60 ppm/V. V_cc2_ tended to increase as the thickness decreased; however, all other conditions except NIT could satisfy the value of 100 ppm/dV^2^ or less when implementing a 2-fF/μm^2^ MIM capacitor.

To investigate the TCC characteristics of the MIM capacitors, the capacitance was measured at the center of the wafers with varying thicknesses and conditions of the Si_x_N_y_ films while incrementally raising the temperature to 25 °C, 50 °C, 75 °C, 100 °C, and 125 °C. The TCC was calculated using Equation (4) [39]:(4)$$\frac{C\left(T\right)-C\left(25\right)}{C\left(25\right)}=T_{cc}T+C$$
where *C*(*T*) is the capacitance under variable temperature, *C*(25) is the capacitance at 25 °C, *T_CC_* is the TCC, and *C* is a constant value.

The TCC serves as an indicator of the degree of change in capacitance in response to temperature variations. Unlike the VCC, the TCC exhibits a linear relationship with temperature. Therefore, when the temperature is plotted on the *X*-axis and the normalized Δ*C* on the *Y*-axis, the slope value corresponds to the TCC value.

As an illustration, a TCC graph for Si_x_N_y_ films fabricated with the NEWMIM condition at thicknesses of 650, 500, and 400 Å is presented in Figure 7. Additionally, the results for the other conditions are presented to depict the variation in the TCC values according to thickness in Figure 8. Except for the NIT condition, the remaining conditions exhibited values below 50 ppm/dT, even as the thickness decreased. Moreover, the TCC values varied according to the condition at identical thicknesses.

As revealed by Table 4, the differences in Si_x_N_y_ conditions are attributed to the N-H/Si-H ratio. To depict the changes in the TCC due to film quality, the N-H/Si-H vs. TCC values are plotted in Figure 9. The results indicate that as the N-H/Si-H ratio increased, the TCC values exhibited an exponential decay trend, confirming that the TCC values are influenced by the quality of the Si_x_N_y_ film.

## 4. Conclusions

An evaluation of the C–V characteristics was conducted for MIM capacitors based on the insulator (Si_x_N_y_) deposition thickness and deposition conditions. The CD values were in the ranges of 0.983–1.1, 1.24–1.4, and 1.57–1.79 fF/µm^2^ for 650, 500, and 400 Å, respectively. Further, the CD increased as the thickness decreased, with variations across different conditions.

At the same thickness, the NIT condition exhibited the highest CD, while the MIM condition showed the lowest. This discrepancy is attributed to the effect of the actual thickness and the difference in the k-value of the Si_x_N_y_ film according to the condition. Additionally, the CD was observed to decrease with increasing capacitor size, possibly due to the influence of fringe capacitance, which increased in proportion to the perimeter/area ratio.

The thickness uniformity of Si_x_N_y_ was found to be the most significant factor affecting capacitance uniformity. Improvements in thickness uniformity can enhance Cp and Cpk on the device side. Across all conditions, a general increase was observed in the VCC as the thickness decreased, although there were some variations between conditions. However, the TCC showed no significant difference with thickness, indicating that the variations were mainly due to the conditions.

In summary, from the perspective of C–V analysis, all conditions, except NIT, demonstrated superior characteristics. Implementing thin Si_x_N_y_ film depositions with stable uniformity using conditions other than NIT could potentially provide MIM capacitors with CD values of less than 100 ppm/dV^2^, aiming for the achievement of 2 fF/µm^2^.

## Figures and Tables

**Figure 1 micromachines-15-01204-f001:**
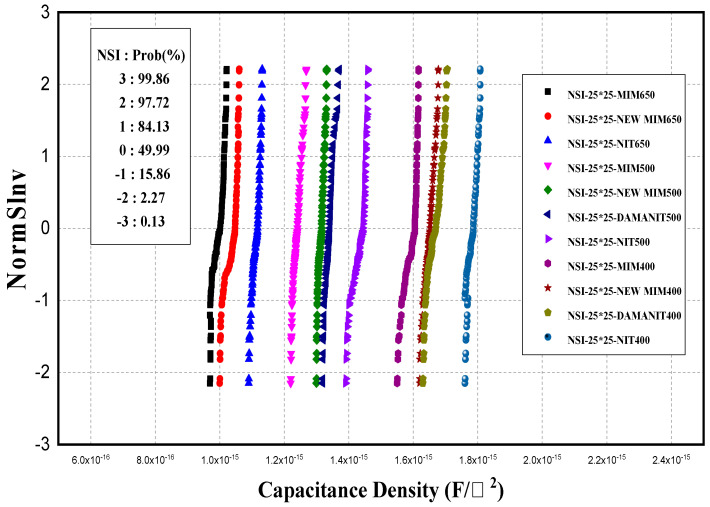
Accumulation curves for CD obtained from different thicknesses and conditions of SixNy films for MIM capacitors.

**Figure 2 micromachines-15-01204-f002:**
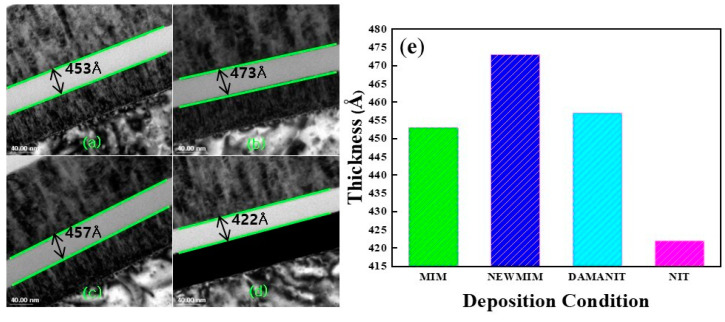
Cross-sectional TEM images of Si_x_N_y_ films with (**a**) MIM, (**b**) NEWMIM, (**c**) DAMANIT, and (**d**) NIT. The thicknesses of the Si_x_N_y_ films are presented in the (**e**) rod graph.

**Figure 3 micromachines-15-01204-f003:**
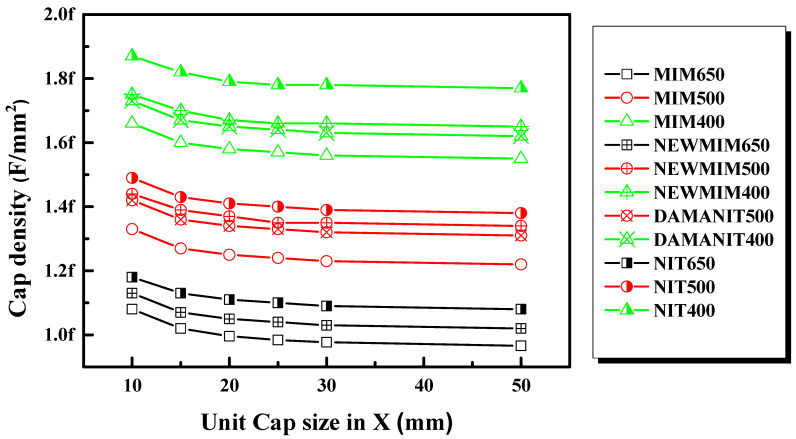
Effect of capacitor size on MIM CD with different Si_x_N_y_ thicknesses and conditions.

**Figure 4 micromachines-15-01204-f004:**
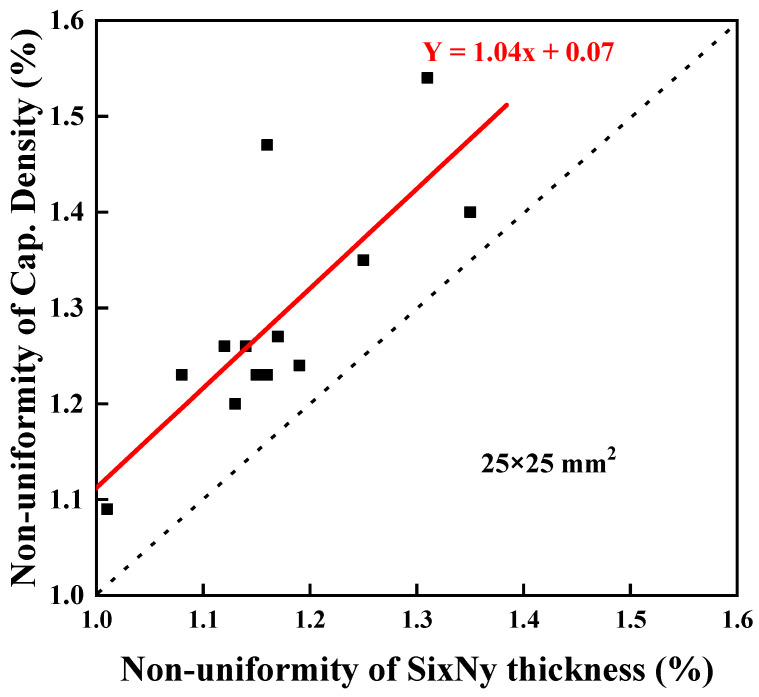
Relationship between nonuniformity of CD and Si_x_N_y_ thickness.

**Figure 5 micromachines-15-01204-f005:**
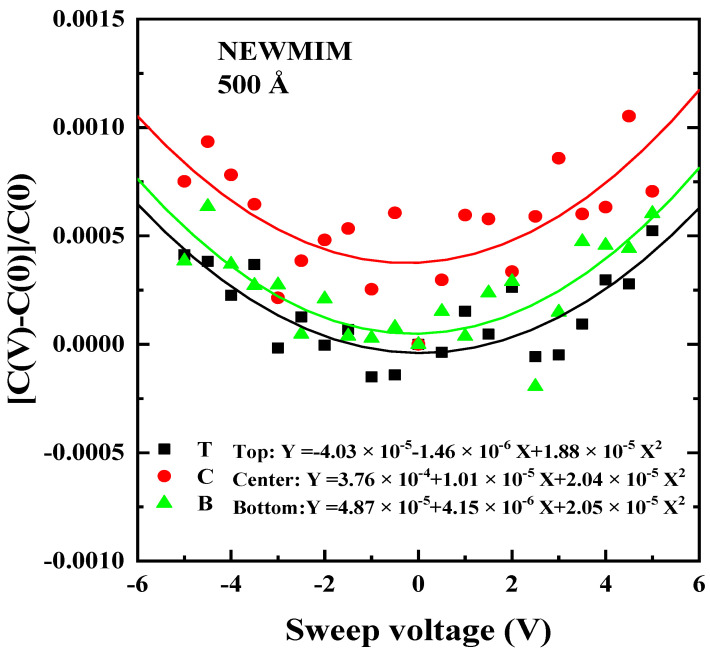
Normalized ΔC vs. voltage at the top, center, and bottom positions. The condition and thickness of the Si_x_N_y_ film are NEWMIM and 500 Ǻ, respectively.

**Figure 6 micromachines-15-01204-f006:**
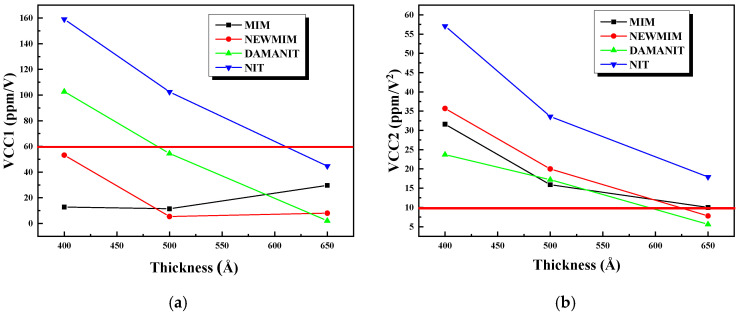
Graphs of (**a**) VCC1 and (**b**) VCC2 vs. thickness with Si_x_N_y_ conditions.

**Figure 7 micromachines-15-01204-f007:**
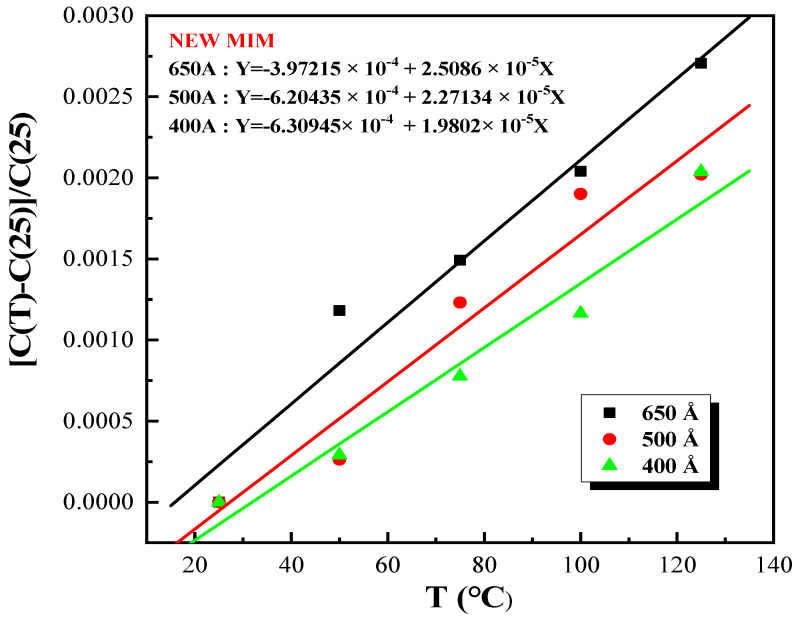
The TCC graph for SixNy films fabricated with the NEWMIM condition at thicknesses of 650, 500, and 400 Å.

**Figure 8 micromachines-15-01204-f008:**
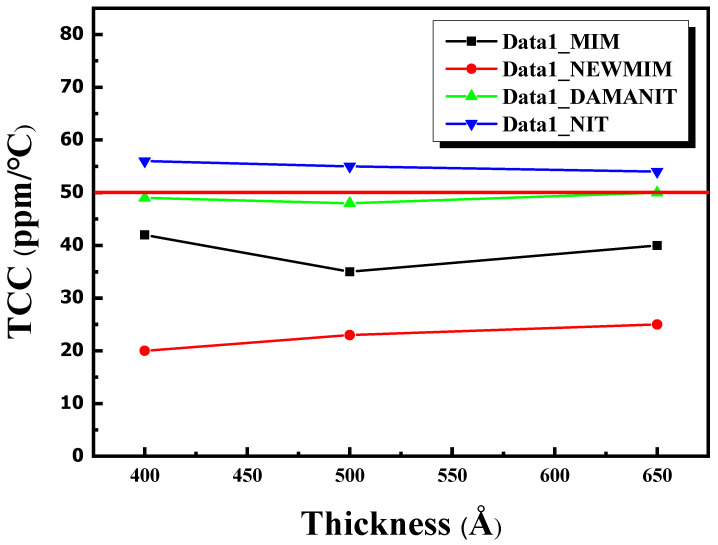
Graph of TCC1 vs. thickness with different Si_x_N_y_ conditions.

**Figure 9 micromachines-15-01204-f009:**
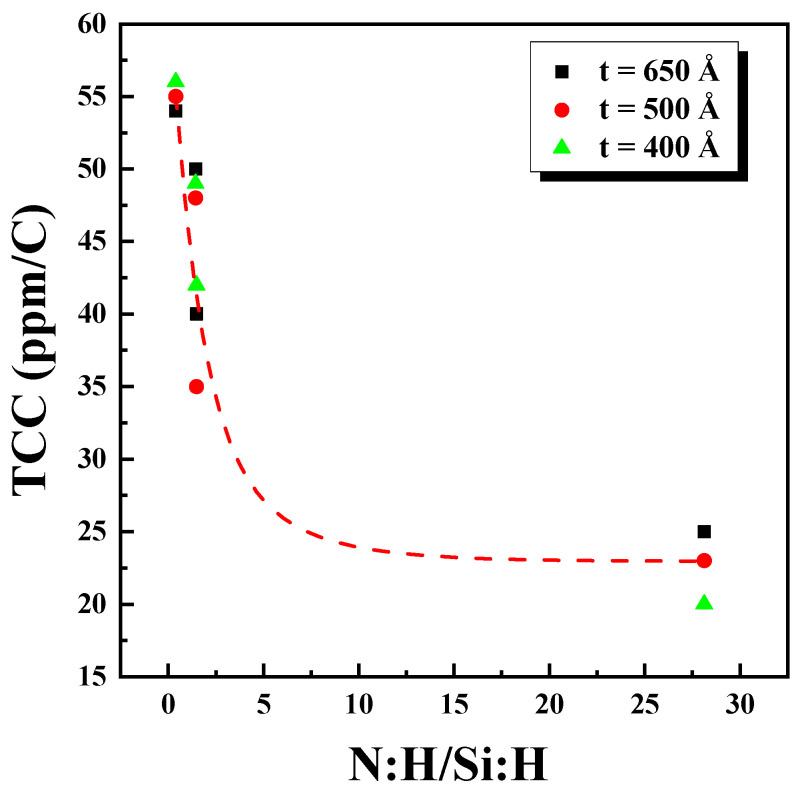
Graph of TCC vs. thickness with varying N:H/Si-H ratio.

**Table 1 micromachines-15-01204-t001:** (**a**) Si_x_N_y_ film properties and (**b**) corresponding process conditions.

**(a)**	**MIM**	**NEW MIM 650**	**DAMA NIT**	**NIT**
Dep. rate	~149 Å/s	~29 Å/s	~59 Å/s	88 Å/s
Within W/F unit (1σ)	1.14%	1.90%	2.34%	2.77%
W/F to W/F unit (1σ)	1.58%	2.21%	1.05%	2.30%
Stress	−2.23 × 10^9^	−1.75 × 10^10^	−2.34 × 10^9^	
H content (N-H: Si-H)	12.7%: 8.5%	22.5%: 0.8%	10.5%: 7.3%	4.4%: 14.8%
N content (N-H/Si-H)	1.49	28.1	1.43	0.30
**(b)**	**MIM 650**	**NEW MIM 650**	**DAMA NIT**	**NIT 650**
Step end control	By time	By time	By time	By time
Maximum step time	4.4 s	22.8 s	11.0 s	
Endpoint selection	No endpoint	No endpoint	No endpoint	No endpoint
Pressure	Servo 4.25 Torr	Servo 4.25 Torr	Servo 4.2 Torr	Servo 4.5 Torr
RF power	690 W	690 W	420 W	425 W
Susc. temperature	400 °C	400 °C	400 °C	400 °C
Susceptor spacing	620 mils	620 mils	550 mils	475 mils
N_2_	3800 sccm	3800 sccm	2500 sccm	4000 sccm
NH_3_	130 sccm	50 sccm	38 sccm	60 sccm
SiH_4_	260 sccm	100 sccm	110 sccm	170 sccm

**Table 2 micromachines-15-01204-t002:** C–V characterization measurement conditions.

Parameter	Setting
Display mode	Cp (parallel capacitor), D (dissipation factor)
Sweep voltage (V)	−5~5
Step (V)	0.5
Oscillation	0.025
Frequency (kHz)	100
Capacitor size (μm^2^)	10 × 10, 15 × 15, 20 × 20, 25 × 25, 30 × 30, 50 × 50
Measurement points	Three points (top, center, and bottom)
Temperature (°C)	25, 50, 75, 100, 125

**Table 3 micromachines-15-01204-t003:** The k-values of the MIM capacitor with Si_x_N_y_ conditions of MIM, NEWMIM, DAMANIT, and NIT.

Deposition Condition	MIM	NEWMIM	DAMANIT	NIT
CD (fF/μm^2^)	1.3256	1.3203	1.3103	1.3606
Thickness (TEM, Å)	453	473	457	422
k-value (ε_0_·ε)	6.00 × 10^−17^	6.25 × 10^−17^	5.99 × 10^−17^	5.74 × 10^−17^

**Table 4 micromachines-15-01204-t004:** Properties of SixNy films according to different conditions.

	MIM 650 DEP	NEW MIM 650 DEP	DAMA NIT 650 DEP	NIT 650 DEP
Deposition rate (Å/s)	~149	~29	~59	88
Within W/F unit (1σ, %)	1.14	1.90	2.34	2.77
W/F to W/F unit (1σ, %)	1.58	2.21	1.05	2.30
Stress	−2.23 × 10^9^	−1.75 × 10^10^	−2.34 × 10^9^	-
H content (N-H:Si-H)	12.7%:8.5%	22.5%:0.8%	10.5%:7.3%	4.4%:14.8%

## Data Availability

The original contributions presented in this study are included in the article. Further inquiries can be directed to the corresponding author.

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
