# Peer review of "Capacitance–Voltage Fluctuation of SixNy-Based Metal–Insulator–Metal Capacitor Due to Silane Surface Treatment"

_micromachines, 2024, doi:10.3390/mi15101204_

Round 1
Reviewer 1 Report
Comments and Suggestions for Authors
In this study, the authors analyze metal–insulator–metal (MIM) capacitors with different thicknesses of a SixNy film (650 Å, 500 Å, and 400 Å) and varying levels of film quality to improve their 10 capacitance density.
They divide the C–V characteristics into two categories: voltage coefficient of capacitance (VCC) and temperature coefficient of capacitance (TCC). General results showed that the capacitance density tended to increase as the thickness of the SixNy film decreased.
The most influential factor for сapacitance uniformity was the thickness uniformity of the SixNy film.
In summary, from the perspective of C–V analysis, all conditions, except NIT, demonstrated superior characteristics.
1) Page 2, lines 49/50 – “ε0 is the permittivity of air”, why air? “free space or vacuum”.
2) Figure 5 shows the normalized values of delta C - in this case, how are the negative values [C(V)-C(0)]/C(0) explained.
3) Is the caption to Figure 7 correct?
4) There is no measurement error (error bar) for the experimental data in Figures 3-9.
Comments on the Quality of English Languagen/a
Author Response
We are grateful for the thorough consideration and scrutiny of our manuscript titled "Capacitance-voltage fluctuation of SixNy based MIM capacitor by Silane surface treatment." The reviewers' comments have helped us better understand the critical issues in this paper. Before responding to each review, please note that we have removed unnecessary references, added missing ones, and removed unnecessary text on the figures (we have not modified any data except for the text).
Comments 1: Page 2, lines 49/50 – “ε0 is the permittivity of air”, why air? “free space or vacuum”.
Response 1: Thanks for pointing this out, we've been made aware of the error in the definition and have fixed it.
Comments 2: Figure 5 shows the normalized values of delta C - in this case, how are the negative values [C(V)-C(0)]/C(0) explained.
Response 2: Thank you for the reviewer's comments. The overall Curve Fitting value converges to 0. The very slightly negative measurement values converge overall to the error range of the measurement equipment. I believe there is no other theoretical meaning, but as you can see in the figure, some very slight negative points appear in the Bottom area of the measurement values at the Top, Center, and Bottom of the wafer. In addition to the previously mentioned measurement equipment hunting, we need to consider the MIM Cap CD variation that appears in the integration processing and the slight variation of the process Uniformity at the Top, Center, and Bottom. In conclusion, considering the Uniformity variation on the wafer and the equipment, we believe that the slight negative values that appear at some points have no significant theoretical meaning.
Comments 3: Is the caption to Figure 7 correct?
Response 3: Thanks to your pointing out, we were able to find our mistake. We've changed the caption appropriately.
Response 4: Thank you for the reviewer's comment. It is true that there should be enough data for statistical significance in normal data, but since the semiconductor equipment and semiconductor process steps performed in the semiconductor fab are very precise equipment and processes that have undergone statistical verification, we did not adopt repeated experiments and performed the wafer process. In addition, since the process progress LOT is very costly when the wafer process is repeated 3-5 times, the data that can display error bars was not displayed as error bars because the equipment process has already undergone statistical verification.
Reviewer 2 Report
Comments and Suggestions for Authors
The comments can be found in the attachment.

Comments on the Quality of English LanguageModerate editing of English language required.
Author Response
[Common Response Comment]
We are grateful for the thorough consideration and scrutiny of our manuscript titled "Capacitance-voltage fluctuation of SixNy based MIM capacitor by Silane surface treatment." The reviewers' comments have helped us better understand the critical issues in this paper. Before responding to each review, please note that we have removed unnecessary references, added missing ones, and removed unnecessary text on the figures (we have not modified any data except for the text).
Comments 1: In this paper, the authors prepared the MIM capacitors based on SixNy films, which are aimed to be applicated in the field of RF integrated circuits (IC). As we know, the SixNy films are conventional dielectric films, not used in the state-of-the-art RF IC. With the miniaturization of IC, the area of MIM capacitors is scaled down. In order to maintain the capacitance, the capacitance density of MIM capacitors has to be correspondingly increased. According to the international technology roadmap of semiconductor (ITRS 2015), the capacitance density is demanded to be larger than 10 fF/um2, together with the quadratic VCC <100 ppm/V2 and leakage current <1E-8 A/cm2@ the operating voltage. Therefore, what is the novelty of this paper? The authors should address this concern.
Response 1: Thank you for the reviewer's comments. I agree with the reviewer's comments on the ITRS roadmap and also agree with the points pointed out from the RF-IC perspective. This paper does not address the viewpoint of increasing the density of the latest MIM capacitance according to the ITRS roadmap. This paper seeks novelty in the analytical approach to see the changes in VCC and TCC according to the nitrogen content. Also, this analysis is a basic research result that is intended to be applied to various fusion sensors rather than to the latest RF-IC chips. Thank you again for your comments.
Comments 2: The title of this paper is Capacitance–voltage fluctuation of SixNy based MIM capacitor by Silane surface treatment. However, the word of silane is present only in the title. There are no details about the silane surface treatment in the experimental section. The authors should give more details about the motivation of using silane surface treatment. Moreover, the detailed procedures about silane surface treatment should be supplemented in the experimental section.
Response 2: Thank you for your thoughtful review. However, we have constructed Si3N4 films with different compositions of N2/NH3/SiH4, which can lead to differences in the ratio of N-H/Si-H bonding inside the thin film. This is described in Tables 1 and 4 of the main text. Based on this, we named this paper as “Capacitance-voltage fluctuation of SixNy based MIM capacitor by Silane surface treatment”.
Comments 3: The leakage current of MIM capacitors is an important parameter for RF application. However, this parameter analysis is lacked in this paper. The authors should supplement the corresponding experiment and analysis to asset the quality of the SixNy films thoroughly.
Response 3: I agree with the reviewer's comments. As the reviewer said, leakage current characteristics are very important characteristics for RF applications. Data related to leakage current characteristics have already been secured, and the data is so vast that it has been submitted as a separate paper. I would like to replace this with references.
Comments 4: In the abstract, the authors show that the capacitance density tended to increase as the thickness of the SixNy film decreased. However, it is a natural conclusion, which can not be regarded as a finding.
Response 4: Thank you for your point. We recognize that the wording is general and not important as a finding, so we have removed it and made some contextual changes to the abstract.
Comments 5: About the SixNy films preparation, the detailed experiment procedures are lacked. The growth method of the SixNy films is CVD or PEALD, which is unclear based on the interpretation in the experimental section. How is the N content of the SixNy films modulated?
Response 5: We appreciate your comment. SixNy Film was deposited using the Plasma Enhanced CVD method of Applied Materials, and the nitrogen content was optimized by controlling the concentration of nitrogen gas during deposition. This deposition method is already commercialized and widely known in semiconductor fabs, so it is not mentioned here. Also, For N content and H content, the values are given in Table 1.
Comments 6:. In Equation 1, ε0 is the vacuum permittivity, not the permittivity of air.
Response 6: Thanks for pointing this out, we've been made aware of the error in the definition and have fixed it.
Reviewer 3 Report
Comments and Suggestions for Authors
In this manuscript, the authors presented a study of C-V fluctuation of SixNy based MIM capacitor by silane surface treatment. This work was interesting. Experimental data were presented and conducted well. Accordingly, I would like to recommend this article. The only suggestion is to provide the scheme for the measured samples.
Author Response
[Common Response Comment]
We are grateful for the thorough consideration and scrutiny of our manuscript titled "Capacitance-voltage fluctuation of SixNy based MIM capacitor by Silane surface treatment." The reviewers' comments have helped us better understand the critical issues in this paper. Before responding to each review, please note that we have removed unnecessary references, added missing ones, and removed unnecessary text on the figures (we have not modified any data except for the text).
Comments 1: In this manuscript, the authors presented a study of C-V fluctuation of SixNy based MIM capacitor by silane surface treatment. This work was interesting. Experimental data were presented and conducted well. Accordingly, I would like to recommend this article. The only suggestion is to provide the scheme for the measured samples.
Response 1: Thank you for your valuable feedback and for recommending our manuscript. We have added a detailed description of the fabrication of the MIM capacitor in Materials and Methods.